# Predicting mitochondrial fission, fusion and depolarisation event locations from a single z-stack

**James G. de Villiers**[ID]**, Rensu P. Theart**[ID]*

Department of Electrical and Electronic Engineering, Stellenbosch University, Stellenbosch, Western Cape, South Africa

* rptheart@sun.ac.za

## Abstract

This paper documents the development of a novel method to predict the occurrence and exact locations of mitochondrial fission, fusion and depolarisation events in three dimensions. This novel implementation of neural networks to predict these events using information encoded only in the morphology of the mitochondria eliminate the need for time-lapse sequences of cells. The ability to predict these morphological mitochondrial events using a single image can not only democratise research but also revolutionise drug trials. The occurrence and location of these events were successfully predicted with a three-dimensional version of the *Pix2Pix* generative adversarial network (GAN) as well as a three-dimensional adversarial segmentation network called the *Vox2Vox* GAN. The *Pix2Pix* GAN predicted the locations of mitochondrial fission, fusion and depolarisation events with accuracies of 35.9%, 33.2% and 4.90%, respectively. Similarly, the *Vox2Vox* GAN achieved accuracies of 37.1%, 37.3% and 7.43%. The accuracies achieved by the networks in this paper are too low for the immediate implementation of these tools in life science research. They do however indicate that the networks have modelled the mitochondrial dynamics to some degree of accuracy and may therefore still be helpful as an indication of where events might occur if time lapse sequences are not available. The prediction of these morphological mitochondrial events have, to our knowledge, never been achieved before in literature. The results from this paper can be used as a baseline for the results obtained by future work.

## Introduction

Mitochondria are highly dynamic organelles present in most mammalian cells and perform the vital biochemical processes of cell respiration and energy production. These organelles are highly networked and capable of rapidly changing form and function to meet the physiological demands of the cell [1]. If mitochondrial DNA (mtDNA) is mutated, it can lead to various common age-related diseases, which includes Alzheimer's disease and certain types of cancer [2–7]. To reduce the risk of mtDNA mutation accumulation, a mitochondrion can undergo dynamic events such as fission, fusion and mitochondrial autophagy (mitophagy) [8, 9].

**Data Availability Statement:** All files are available on Figshare at the following urls: https://figshare.com/articles/figure/Dirty_Dataset_Only_Events_Validation_Set/19726000 https://figshare.com/articles/figure/Dirty_Dataset_Only_Events_

Training_Set/19725997 https://figshare.com/
articles/figure/Dirty_Dataset_Events_On_
Mitochondrial_Strands_Validation_Set/19725994
https://figshare.com/articles/figure/Dirty_Dataset_
Events_On_Mitochondrial_Strands_Training_Set/
19725991 https://figshare.com/articles/figure/
Clean_Dataset_Only_Events_Validation_Set/
19725988 https://figshare.com/articles/figure/
Clean_Dataset_Only_Events_Training_Set/
19725982 https://figshare.com/articles/figure/
Clean_Dataset_Events_On_Mitochondrial_
Strands_Validation_Set/19725967 https://figshare.
com/articles/figure/Clean_Dataset_Events_On_
Mitochondrial_Strands_Training_Set/19725958.

**Funding:** - JG de Villiers was received a
scholarship from the National Research Foundation
(NRF). - Grant number: MND200423516047 -
National Research Foundation (NRF) - URL: https://
www.nrf.ac.za/ - No, they played no role in the
study.

**Competing interests:** The co-author, Dr RP Theart,
is the author and creator of the Mitochondrial Event
Localiser, which was used to generate the ground-
truth data for this project. His research is also
published in PLOS ONE (https://doi.org/10.1371/
journal.pone.0229634). This does not alter our
adherence to PLOS ONE policies on sharing data
and materials.

Fission is the division of a mitochondrion into two or more distinct daughter organelles and fusion is the inverse of this process. The fission and fusion processes play vital roles in the quality control of cells and mitochondrial health, which is crucial for cellular homeostasis [10].

Adenosine triphosphate (ATP) is produced in the mitochondria through a process called oxidative phosphorylation, for which the mitochondria must have constant membrane potentials [11]. Negative by-products of this process are reactive oxygen species (ROS), which are free radicals capable of damaging the organelles and possibly leading to the mutation of its DNA [12]. The mitochondrial respiratory chain is the main site of cellular ROS production, while mitochondria are also highly sensitive to, and affected by, ROS toxicity [13]. ROS induced ROS release (RIRR) has been described as a positive feedback mechanism that is involved in the interaction between ROS and the mitochondria where a surge of mitochondrial ROS is induced by ROS, subsequently causing a reduction in the mitochondrial membrane potential [14]. The organelle can undergo membrane depolarisation (changing the polarity of its membranes) to reduce the production of these reactive oxygen species [15, 16]. If depolarisation is sustained for prolonged periods, it can lead to cell death [17]. For this reason mitochondrial depolarisation is required for mitochondrial quality control and removal of dysfunctional mitochondria through mitophagy.

Until recently, the occurrence and locations of these events were determined through manual visual inspection by comparing two frames of a time-lapse sequence. Recently, an automatic method for comparing two frames of a time-lapse sequence to localise these events has been developed [18]. This automatic method was used to generate the training data for the neural networks discussed in this paper. These networks use a single frame of a time-lapse sequence as input and makes predictions using information encoded in the morphology of the mitochondria.

Mitochondrial event predictions using neural networks enable insight into the health of cells from a single z-stack, thereby eliminating the need to capture time-lapse sequences. The removal of time-lapse sequences from the analysis process enables the use of fixed cells rather than live cells while still obtaining similar analysis insights. Furthermore, neural networks democratise this type of research by providing researchers who have no access to the equipment necessary to obtain time-lapse sequences with the opportunity to conduct similar studies.

The ability to predict the locations of these events on a z-stack image using only a single image and information encoded only in the morphology of the mitochondria implied that generative networks were needed. For this reason generative neural networks were used. These networks were trained using pairs of input and desired output images. To our knowledge the use of generative networks to predict the locations of mitochondrial events has never been done before.

## Related work

To our knowledge, no other methods capable of predicting these events exist and as a consequence this section discusses the automatic method that was is used for localising mitochondrial fission, fusion and depolarisation events by comparing two frames of a time-lapse sequence, which was used to generate the ground truth data for the neural networks discussed in this paper. This section also discusses the most prominent generative neural networks used in industry.

### Mitochondrial event localiser (MEL)

The localisation of mitochondrial fission, fusion and depolarisation events, especially in three-dimensions, has, until, recently been done manually. The mitochondrial event localiser

(MEL), developed by Theart *et al.* [18], is an automatic, deterministic and high-throughput analysis system. MEL uses two sequential frames of a time-lapse sequence of fluorescence microscopy z-stack images that was stained for mitochondria (with TMRE) as input and provides the three-dimensional locations of mitochondrial fission, fusion and depolarisation events as output.

MEL is, to our knowledge, the only method currently available to automatically localise mitochondrial events. The method is not perfect, and does introduce false-positive events during localisation. For this reason the outputs of MEL are manually evaluated by the user by comparing the localised events to the relevant frames of the time-lapse sequence using a tool developed by the author of MEL, which is shown in S1 Fig. This process allows for the removal of false-positive event localisations, but does not solve the problem of "missed" events.

For this reason the dataset created by MEL cannot be viewed as the absolute ground-truth, but purely a good approximation of it. It is however not feasible to create such a training dataset by hand since it would take to long to complete and not necessarily guarantee an increase in accuracy.

An adapted version of MEL was used in order generate the labelled data for this paper.

## Generative neural networks

The *Pix2Pix* GAN was developed specifically for image-to-image translations and was first introduced in 2016 by Isola, *et al.* [19]. In order to train the network, both input and target output images are passed to the network. The generator network encodes the input image to a lower dimensional latent vector and then tries to generate an output image that is indistinguishable from the target image. The discriminator classifies its input images as real or fake.

The *Vox2Vox* GAN is a three-dimensional adversarial image segmentation network [20]. It is similar in architecture and principle to the *Pix2Pix* GAN, but differs with an alteration to the bottle-neck. The *Vox2Vox* generator does not encode the input z-stack to a single row latent vector, but instead to a three-dimensional array. This array is then passed to a bottle-neck with four *Res-Net* style encoder blocks and then decoded to the output image.

## Materials and methods

For this project, U-118MG mammalian cells were purchased from the *American Type Culture Collection*. These cells were supplemented with *Dulbecco's Modified Eagles Medium* (catalogue number 41965062, Life Technologies), 1% penicillin-streptomycin (catalogue number 15140122, Life Technologies) and 10% fetal bovine serum (catalogue number BC/50615-HI, Life Sciences). They were incubated in a *SL SHEL LAB $CO_2$* humidified incubator in the presence of 5% $CO_2$ at 37 C.

To allow for visualisation of the mitochondrial network, the U-118MG cells were seeded in 8-chamber *Nunc®️ Lab-Tek®️ II* dishes and incubated with 100 nM tetramethylrhodamine-ethyl ester (TMRE)(catalogue number 87917, Sigma-Aldrich) for 5 minutes in the presence of 5% $CO_2$ at 37 C. TMRE is used widely when assessing mitochondrial dysfunction and has a good signal-to-noise ratio [21]. Additionally, since TMRE is a membrane potential dependent chemical dye it allows us to create the ground truth data for mitochondrial depolarisation which MEL is able to detect.

### Microscopy

A Carl Zeiss Elyra PS1 microscope was used to conduct live cell confocal microscopy. Z-stacks images were acquired with a 0.5 $\mu$m step width, Plan-Apochromat 100x/1.46 oil DIC M27 objective, GaAsP detector and a 561 nm laser as illumination source. A time-lapse sequence of

mitochondrial dynamics was recorded with 10 s intervals for 30 cycles. The laser was used at a low power setting to limit photo-bleaching and photo-oxidative stress during the acquisition period.

## Mitochondrial event localiser training data generation

MEL was used to localise mitochondrial fission, fusion and depolarisation events for 44 three-dimensional distinct fluorescence microscopy z-stack time-lapse sequences of healthy, untreated, cells. The parameters proposed by Theart *et al.* [18] were used to create the dataset for the training of the neural network.

A review tool was developed by Theart *et al.* [18], which was used to validate the legitimacy of the events localised by MEL by manually comparing the relevant frames of the time-lapse sequence to the localised events. False-positive events were removed form the dataset through this process.

After the removal of all of the unsatisfactory samples, a dataset containing 944 z-stacks remained. From this, 94 z-stacks (approximately 10%) were used as validation z-stacks and the remaining 850 z-stacks as training z-stacks.

In order to create a subset of the best representative samples, these 944 z-stacks were once again manually evaluated by comparing their relevant frames in the time-lapse sequence to the validated events in order to determine the most appropriate z-stacks that should be used for training. A total of 179 z-stacks were selected as the best samples for training, which was divided into 160 training and 19 validation z-stacks. For the remainder of this paper, the data-set containing all 944 z-stacks will be referred to as the complete dataset and the subset containing 179 z-stacks as the clean dataset.

## Dataset deviation from absolute ground-truth

The MEL method uses various image pre-processing steps, which, however vital for the process, leads to localisation errors such as false-positive event localisations or the omission of true-positive events. The choice of pre-processing parameters could have an effect on the binarisation, which would affect the localisation accuracy of MEL. It is however not possible to eliminate the false-positives or omission of true-positives and as a consequence when choosing the MEL pre-processing parameters the aim is to find the best compromise of these two.

A common cause of localisation errors was the binarisation of low intensity bridges which were created during imaging between two structures that were in close proximity. These low intensity bridges were incorrectly binarised by the thresholding process, consequently creating one large structure instead of two separate structures. MEL only uses these binarised z-stacks, and not intensity z-stacks, to localise the mitochondrial events. S2 Fig shows a good example in which MEL would have incorrectly localised a fusion event due to the thresholding of a low intensity bridge between structures.

To ensure that the neural networks are not adversely affected by this limitation of binarised data, the neural networks, discussed in this paper, received the deconvolved z-stacks as input, rather than the thresholded z-stacks. It was expected that the networks will predict events that were not localised by MEL, because the network is expected to also discern between mitochondrial structures and the background. This would imply that the low intensity bridges, which were created by the thresholding process in MEL, would not be passed as input to the neural network. In contrast, the MEL method received thresholded z-stacks with a clear boundary between mitochondrial structures and the background, which limited its localisation performance. It was presumed that the neural network would perform better than a thresholding method when discerning between foreground and background pixels.

This lack of accurate ground-truth data complicated the optimisation of the neural networks, because the validation loss cannot be used as the metric for determining the optimal point of the network. Due to the use of the deconvolved z-stacks as input, it was likely that the neural network would predict more (true-positive) events than which were localised by MEL. This could lead to a high validation loss but a more accurate representation of the mitochondrial events.

### Location based weighted mean absolute error loss function

The nature of the training data meant that standard loss functions would be unsuitable for our application. The mean absolute error (MAE) loss was modified by adding a penalty factor to the voxels in the generator output that did not contain the events localised by MEL. To our knowledge, adapting the standard MAE loss has not been documented before. This modified MAE loss was named the event location penalised MAE loss (ELP-MAE).

To do this, MEL was first altered to provide two z-stacks as output. One z-stack containing only the localised events, and the second the localised events superimposed on the first input frame of the time-lapse sequence. The two output z-stacks of MEL are illustrated in Fig 1.

The voxel values of the events localised by MEL (Fig 1A) were all assigned a value of 1 and the surrounding background was assigned a value of 0. This array is henceforth referred to as the thresholded events z-stack.

The ELP-MAE is designed to receive three z-stacks, namely the ground-truth, the generator output and the thresholded events z-stack are passed as input to the loss function. However, the implementation of loss functions in the *Keras Python* library allows for only two inputs, namely the predicted value ($y_{predicted}$) and the ground-truth value ($y_{true}$).

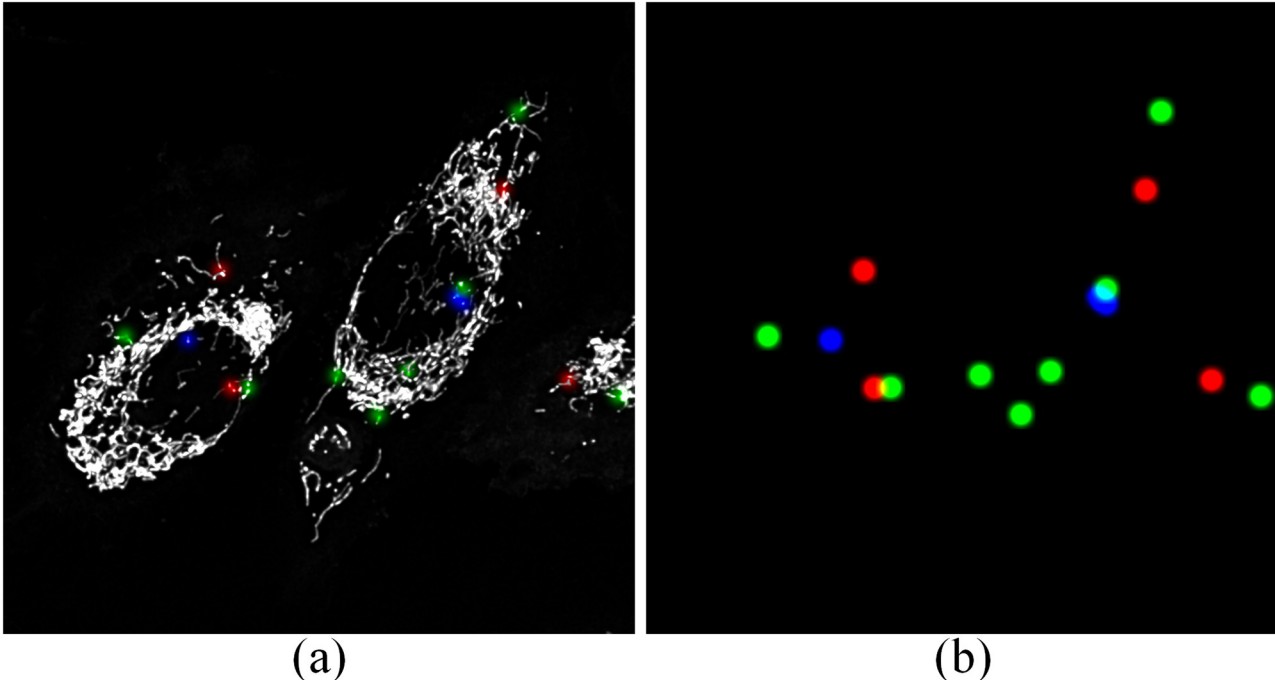

(a)　　　　　　　　　　　　　(b)

**Fig 1. Maximum intensity projections of the two output z-stacks, which are generated by MEL.** (A) The first z-stack contains the localised mitochondrial events that were superimposed on the mitochondrial structures, and (B) the second z-stack contains only the localised events. The contrast and saturation of the z-stacks where enhanced for visual clarity.

In order to overcome this restriction of the *Keras* library, the thresholded events z-stack was concatenated with the ground-truth z-stack (shown in Fig 1B) to create a $2 \times 8 \times 128 \times 128 \times 3$ array, which will henceforth be referred to as the stacked array. For the implementation of the ELP-MAE, the stacked array is passed as the $y_{true}$ input, whilst the output of the generator remained the $y_{predicted}$ input to the loss function.

Fig 2 provides an illustration of the ELP-MAE loss function implementation within the *Keras Python* library on a $2 \times 2$ pixel example image. The black pixels indicates background pixels with a value of 0, and white pixels are the foreground pixels with a value of 1. The red pixel indicates a mitochondrial event, also with a pixel value of 1.

Within the function that implements the ELP-MAE, the $2 \times 8 \times 128 \times 128 \times 3$ stacked array is split into two $8 \times 128 \times 128 \times 3$ z-stacks. The one z-stack is the ground truth ($y_{true}$) and the other one the thresholded events z-stack. The absolute error between the ground-truth and the generator output ($y_{predicted}$) is calculated and the thresholded events z-stack is multiplied with a user-defined penalty value (100 in this example), after which a value of 1 is added. This is done to prevent the elimination (multiplication with zero) of error values in areas where mitochondrial events are not present in the thresholded events array. If this addition of 1 is not implemented, only the absolute errors in the areas containing mitochondrial events will be used to update the parameters of the GAN. The network would perceive that the surrounding regions have an error value of zero and therefore would not contribute to the training of the generator. This would result in an untrainable model.

The final step of the ELP-MAE loss function is to multiply the penalised thresholded events z-stack with the absolute error, after which the mean of the voxel values is calculated.

During the training iterations it was found that increasing the penalty factor led to increased detail in the predicted events, however, this also caused an increase in the creation of unwanted random noise artefacts by the generator. These noise artefacts were reduced with

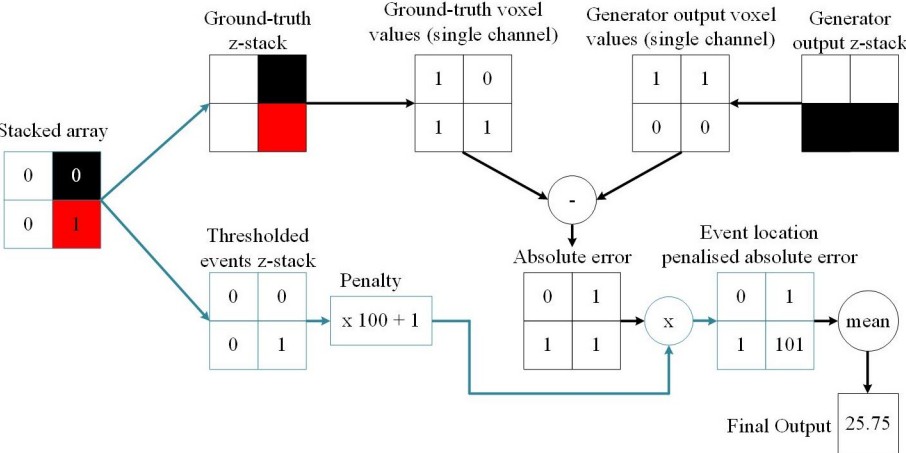

**Fig 2. A diagram of the ELP-MAE loss function that was used to train generative neural networks.** The ELP-MAE loss function penalises areas where mitochondrial events should have been predicted in the generator output. The function first splits the stacked array into the ground-truth and the binarised events z-stacks. The absolute error between the ground-truth and generator output z-stacks is calculated. The thresholded events z-stack that contains only the localised events, is multiplied with a user-defined penalty value (e.g. 100). A value of 1 is then added to all elements in the array to prevent the elimination (multiplication with zero) of error values in areas where mitochondrial events are not present. The final output of the loss function is the mean of the error values after the penalty factor was applied. This is a much larger value than what would have been obtained with the MAE loss. The variations of the ELP-MAE on the standard MAE loss are shown with blue lines.

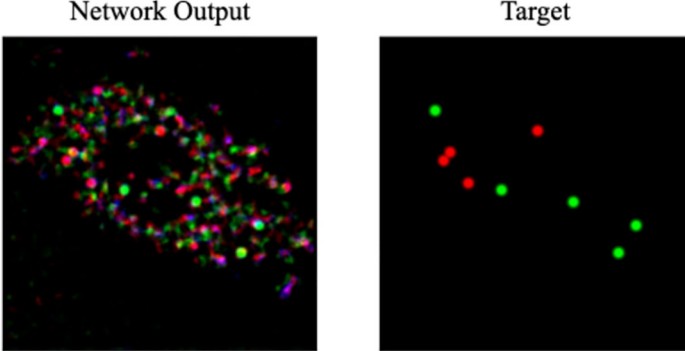

**Fig 3. The output of the generator network using the ELP-MAE loss.** The output image can be compared to the target image to see the accuracy of the network output. A negative by-product of the penalty factor that is used in the ELP-MAE loss function are random noise artefacts.

noise-reduction training methods, which will be discussed later. Fig 3 shows the noise artefacts that were generated by the generator for one of the training images.

It was also found that if the penalty factor was too small, the neural network was unable to predict the locations of mitochondrial events. On the other hand, if the penalty was too large, excessive noise artefacts were generated. The penalty factors that were used were iteratively determined during training.

A pertinent shortcoming of the neural network training, which is discussed in this paper, is the lack of an accurate method to validate the accuracy of the predictions made by these networks. The dataset generated by MEL cannot be viewed as the absolute ground-truth, since localisation errors due to image pre-processing and spatial movement of the mitochondria affects the accuracy of its localisations. For this reason, the ELP-MAE loss for the validation dataset can not be monitored during training to determine when the network has reached an optimal point. Visual inspection of the generator's outputs after every few iterations was the only viable method of monitoring the accuracy. This complicated the optimisation of the neural network, because alterations leading to slight improvements in training were difficult to notice through visual inspection.

## Mitochondrial event location prediction in three dimensions

Two methods of predicting the locations of mitochondrial fission, fusion and depolarisation events in three dimensions were investigated. The first method was with the use of the *Pix2Pix* GAN and the second with the *Vox2Vox* GAN. These networks received a single z-stack from a time-lapse sequence as input and were tasked with producing z-stacks containing the event locations.

**Three-dimensional Pix2Pix GAN generator architecture.** The generator network that was used in the *Pix2Pix* GAN is similar to the standard architecture, as described by Isola et al., [19]. It contains an encoder and a decoder network with skip connections between them, as illustrated by Fig 4.

The encoding region of the generator was constructed from six encoder blocks, each containing a three-dimensional convolution layer with kernels of size 4, followed by an instance normalisation and a Leaky ReLU activation layer. Normalisation was not applied in the first encoding block in order to preserve the semantic information of the input z-stack, which is often lost in the encoder. The output of each encoding block is passed across the network via a

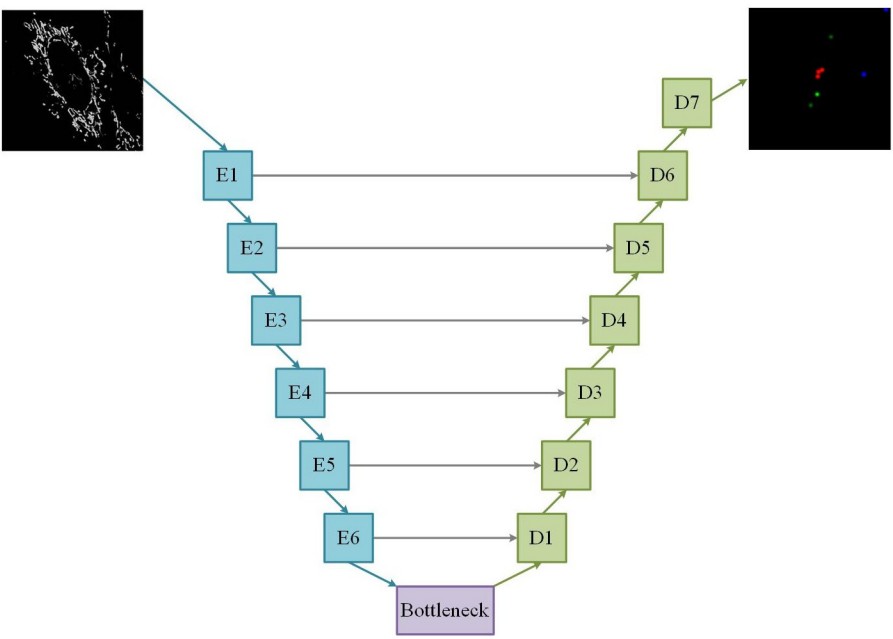

**Fig 4. The *Pix2Pix* GAN generator architecture.** The generator is constructed from six encoder blocks (*E*) and seven decoder blocks (*D*). A single frame from a time-lapse sequence is encoded to a lower-dimensional latent vector (bottle-neck), which is decoded to an z-stack containing the locations of mitochondrial events. The horizontal arrows represent skip connections.

skip connection to a corresponding decoding block. The stride lengths and kernel sizes used in each encoding block are listed in Table 1

The final encoding block is followed by the bottle-neck, which contains a single convolutional layer and ReLU activation layer. Normalisation was not applied to the bottle-neck because it would zero the activations, which will cause the network to skip the bottle-neck [19].

The decoder blocks were constructed from three-dimensional up-sampling layers, followed by three-dimensional convolutional layers with kernels of size 4 and stride lengths of 1 in all three directions. After this, instance normalisation, the ReLU activation function and dropout with a rate of 50% were applied. The output of the decoder block was concatenated along the colour axis with its corresponding skip connection. Dropout layers were only used in the first four decoder blocks, as was suggested by Isola *et al.* [19]. Normalisation was not applied in the final decoding block and the hyperbolic tangent activation function was used to generate

**Table 1. The number of filters and stride lengths that were used in the three-dimensional *Pix2Pix* GAN's encoding blocks.**

| Encoding block | Number of filters | Stride lengths (z,x,y) |
| --- | --- | --- |
| *E1* | 64 | (1,2,2) |
| *E2* | 128 | (1,2,2) |
| *E3* | 256 | (1,2,2) |
| *E4* | 256 | (2,2,2) |
| *E5* | 512 | (2,2,2) |
| *E6* | 512 | (2,2,2) |
| *bottle − neck* | 512 | (1,2,2) |

**Table 2. The number of filters and up-sampling scales that were used in the three-dimensional *Pix2Pix* GAN's decoding blocks.**

| Decoding block | Number of filters | Up-sampling scale (z,x,y) |
|---|---|---|
| *D*1 | 512 | (1,2,2) |
| *D*2 | 512 | (2,2,2) |
| *D*3 | 256 | (2,2,2) |
| *D*4 | 256 | (2,2,2) |
| *D*5 | 128 | (1,2,2) |
| *D*6 | 64 | (1,2,2) |
| *D*7 | 3 | (1,2,2) |

output z-stacks with voxel values in a range of -1 to 1, which is the accepted norm for GANs. The parameters that were used for each decoder block in the generator architecture are listed in Table 2. The additional decoder block in the generator architecture is used to scale the output of the previous decoder block to the desired dimensions of the target z-stack.

The number of filters used for the convolutional layers, and consequently the depth of the neural network, was maximised for the available VRAM of 12 GB.

**Three-dimensional Pix2Pix GAN discriminator architecture.** The architecture of the discriminator, as discussed in Isola *et al.* [19], was modified to work for three-dimensional z-stacks. The concatenated input and target/generator output z-stacks were used as input to the discriminator network. The network was constructed from 5 encoding blocks, which reduced the input z-stack to a $1 \times 8 \times 8 \times 1$ patch of probability values. Each voxel in the $1 \times 8 \times 8 \times 1$ patch output of the discriminator relates to a $8 \times 94 \times 94$ cube of voxels in the input z-stack. This relation is called the receptive field of the generator. The architecture of the discriminator is shown in Fig 5.

The discriminator encoding blocks were constructed from three-dimensional convolutional layers with kernels of size 4. The convolutional layers were followed by Leaky ReLU activation and instance normalisation layers. A slope of 0.2 was used for the Leaky ReLU activation function. In order to preserve semantic information, such as the colours of the input z-stacks, normalisation was not applied in the first block, as was suggested by Isola *et al.* [19]. In order for the discriminator to assign values close to or equal to 0 for fake z-stacks, and 1 for real z-stacks, the output of the last encoding block was passed to a sigmoid activation function. The binary cross-entropy loss was used as the loss function for the discriminator. The parameters that were used for each of the discriminator encoding blocks are listed in Table 3.

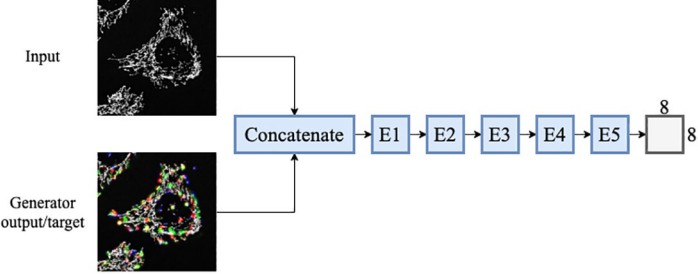

**Fig 5. The discriminator architecture of the three-dimensional *Pix2Pix* GAN.** The discriminator receives and concatenates both the input frame and the generator output/target z-stack as input.

**Table 3. The number of filters and stride lengths that were used in the three-dimensional *Pix2Pix* GAN's discriminator encoding blocks.**

| Encoding block | Number of filters | Stride lengths (z,x,y) |
|---|---|---|
| E1 | 64 | (2,2,2) |
| E2 | 128 | (2,2,2) |
| E3 | 256 | (2,2,2) |
| E4 | 512 | (1,2,2) |
| E5 | 1 | (1,1,1) |

The clean dataset, which contains 179 z-stacks as discussed in a previous section, was used to train the network with a large penalty value in the ELP-MAE loss function. This created random noise artefacts in the output z-stacks, which were reduced by continuing the training of the network using the complete dataset with a low penalty value. This continuation of training with the complete dataset will henceforth be referred to as noise reduction training.

The training of the *Pix2Pix* GAN with the clean dataset was deemed to be complete after 545 epochs with 16 mini-batches of size 10 and a ELP-MAE loss penalty value of 50 000. This penalty value was iteratively determined to be large enough to ensure the correct training of the generator, whilst not generating excessive amounts of noise.

Training with the clean dataset was deemed to be sufficient after 545 epochs, because the network predicted mitochondrial events with bright, round kernels on the validation z-stacks and the generator training loss remained constant.

Training of the network was continued using the complete dataset containing 944 samples, for an additional 25 epochs, with 85 mini-batches of size 10, and a penalty value of 500. After the 25 epochs of noise reduction training there was no further reduction in noise artefacts and the brightness of the predicted events began to fade. The penalty value of 500 was small enough to allow for a significant reduction of the noise artefacts, whilst not completely eliminating predicted events. Fig 6 shows the difference in results obtained on one of the validation z-stacks, before and after noise reduction training with the complete dataset.

## Vox2Vox generative adversarial netowrk

The *Vox2Vox* GAN was specifically designed for three-dimensional adversarial z-stack segmentation tasks. Similar to the *Pix2Pix* GAN, the *Vox2Vox* received a single frame of a time-lapse sequence as input. The architecture and the training of the *Vox2Vox* GAN will be discussed in this section.

**Vox2Vox generative adversarial network generator architecture.** The *Vox2Vox* GAN was constructed according to the specifications discussed in Cirillo *et al.* [20]. The architecture of the generator consists of an encoder, bottle-neck, decoder and skip connections, as illustrated in Fig 7. The generator architecture of the *Vox2Vox* GAN is shallower than that of the *Pix2Pix* GAN. This was due to an increase in the trainable parameters of the network (329 546 755 for the *Vox2Vox* GAN compared to 129 546 755 for the *Pix2Pix* GAN), as a consequence of the short skip connections in die bottle-neck of the *Vox2Vox* generator. Theoretically, a deeper model would preform better and should be investigated with improved hardware.

The encoder was constructed from four encoding blocks similar to those that were used for the *Pix2Pix* GAN. This was followed by the bottle-neck region, which contained four bottle-neck blocks with *Res-Net* style short skip connections. These blocks were constructed from three-dimensional convolutional layers with kernels of size 4, followed by instance normalisation, Leaky ReLU activation and dropout layers.

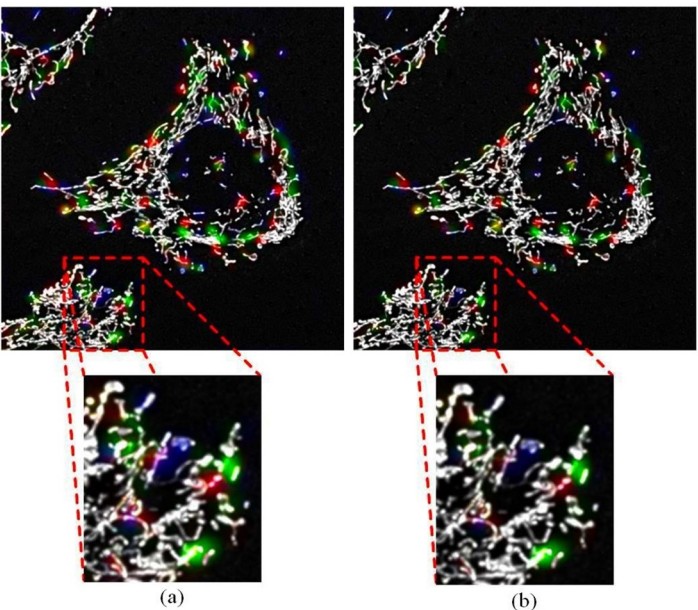

**Fig 6. The output of the three-dimensional _Pix2Pix_ GAN after training.** (A) The output initial training with the clean dataset for 545 epochs and (B) then after training with the complete dataset for an additional 25 epochs to reduce noise artefacts. After this noise reduction training process, the random colour fragments were reduced, as illustrated within the enlarged red rectangles. During validation, the two green kernels on the right of the enlarged z-stacks are counted as predicted fusion events.

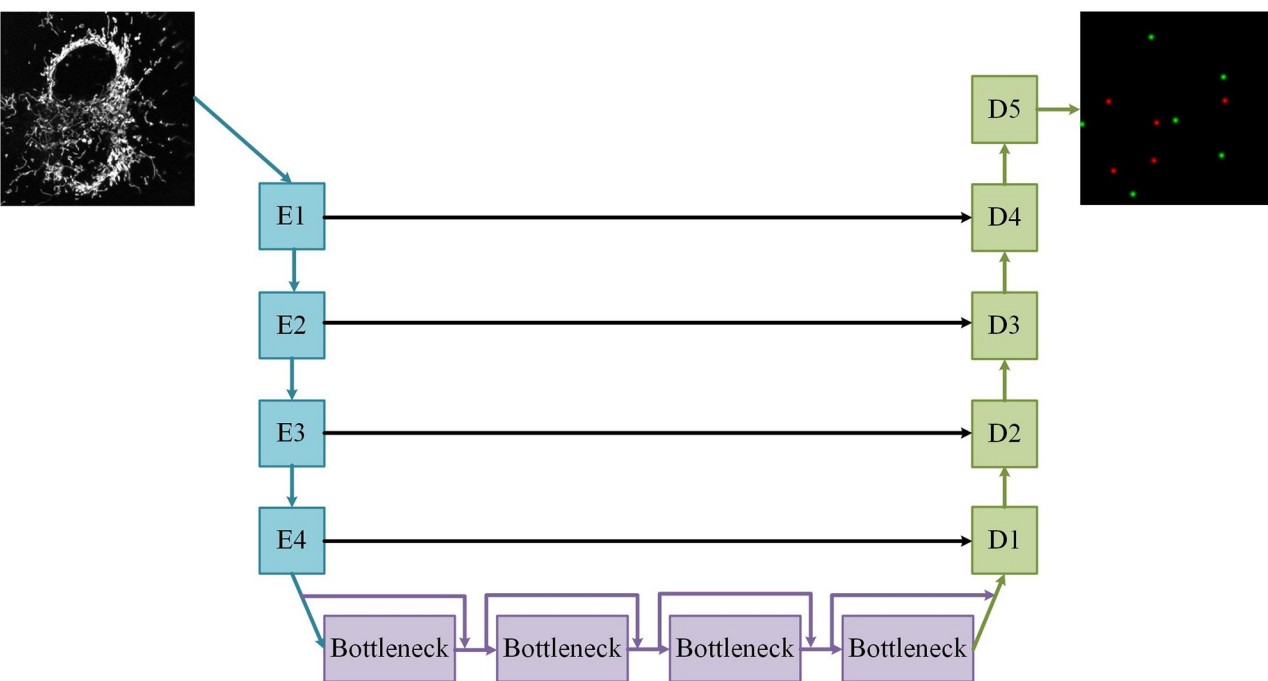

**Fig 7. The _Vox2Vox_ GAN generator architecture.** The generator is constructed from four encoder blocks (_E_) and five decoder blocks (_D_), connected with four bottle-neck blocks. The black arrows represent the skip connections and the purple arrows the short _Res-Net_ style short skip connections. A single input frame is used to predict the locations of mitochondrial fission, fusion and depolarisation events on a black background.

**Table 4. The number of filters and stride lengths that were used in the _Vox2Vox_ GAN's encoding blocks.**

| Encoding block | Number of filters | Stride length (z,x,y) |
|---|---|---|
| _E_1 | 64 | (2,2,2) |
| _E_2 | 128 | (2,2,2) |
| _E_3 | 256 | (2,2,2) |
| _E_4 | 512 | (1,2,2) |
| _bottle – neck_ | 512 | (1,1,1) |
| _bottle – neck_ | 512 | (1,1,1) |
| _bottle – neck_ | 512 | (1,1,1) |
| _bottle – neck_ | 512 | (1,1,1) |

All Leaky ReLU functions had a slope of 0.2 and the dropout was applied at a rate of 20%, as suggested by Cirillo _et al._ [20]. The parameters that were used in the encoding and bottle-neck blocks are listed in Table 4.

The _Vox2Vox_ generator's decoding region was constructed from five decoding blocks with similar architectures to those that were used for the _Pix2Pix_ GAN. A fifth decoding block was added used scale the output of fourth decoding block to the dimensions of the target output image.

The decoder blocks were constructed from a three-dimensional up-sampling layer, followed by a convolutional layer with kernels of size 4 and strides lengths of 1 in all directions. The ReLU activation and instance normalisation layers were used in these blocks. Similar to the _Pix2Pix_ GAN, and as dictated by convention [22], normalisation was not applied in the last decoder block and the hyperbolic tangent activation function was used to normalise the voxel values in a range from -1 to 1. The parameters that were used for the decoding blocks of the _Vox2Vox_ GAN are listed in Table 5. The listed output sizes are the dimensions after the decoder block output was concatenated with the skip connections.

The architecture of the _Vox2Vox_ discriminator is identical to that of the _Pix2Pix_ GAN as was shown in Fig 5.

**Vox2Vox generative adversarial network training.** Training of the _Vox2Vox_ GAN was done using the same principles as for the _Pix2Pix_ GAN. The network was trained for 514 epochs with an ELP-MAE penalty of 50 000, using the clean dataset, which consisted of 40 mini-batches, each with a batch size of 4.

After 514 epochs of training, the generator loss stabilised, and events were predicted with bright kernels. Similar to the _Pix2Pix_ GAN, the penalty value was determined iteratively to be large enough for the sufficient penalisation of the areas containing mitochondrial events. This ensured the optimal training of the generator, whilst not generating excessive amounts of noise.

To reduce noise, the network was trained using the complete dataset for an additional 26 epochs with an ELP-MAE penalty of 500. For this training, 170 mini-batches with a batch size

**Table 5. The number of filters and up-sampling scales that were used in the _Vox2Vox_ GAN's decoding blocks.**

| Decoding block | Number of filters | Up-sampling scale (z,x,y) |
|---|---|---|
| _D_1 | 512 | (1,1,1) |
| _D_2 | 512 | (1,2,2) |
| _D_3 | 256 | (2,2,2) |
| _D_4 | 256 | (2,2,2) |
| _D_5 | 128 | (2,2,2) |

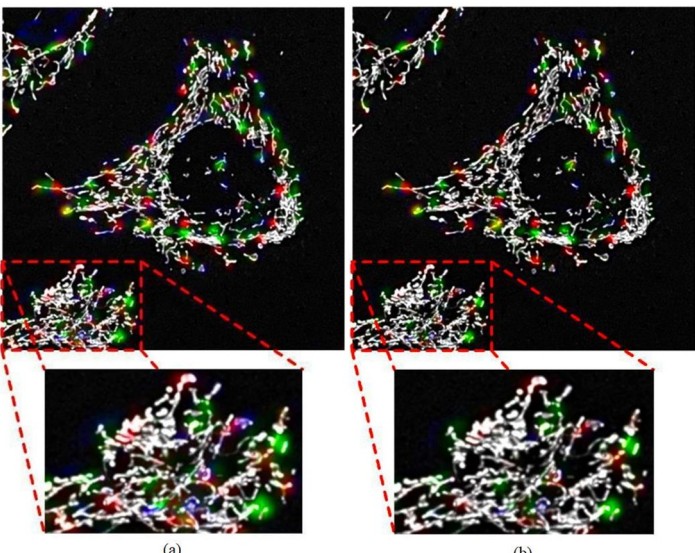

**Fig 8. The output of the *Vox2Vox* GAN after training.** (A) The output after training with the clean dataset for 514 epochs and (B) after training with the complete dataset for an additional 26 epochs to reduce noise artefacts. The reduction in random noise artefacts is illustrated within the enlarged view of the red rectangles, which was achieved by using the complete dataset for noise reduction training. More than 26 epochs of noise reduction training did not further improve the outputs of the generator. Some noise fragments remained, however, only the two predicted fusion events (green kernels) on the right of the enlarged z-stack were regarded as predicted events during validation.

of 5 were used. After 26 epochs the validation loss began increasing, which indicated over-fitting, and no further noise reduction was observed. The output of the network after 514 epochs and 540 epochs of training, is shown in Fig 8 for one of the validation z-stacks.

## Results and discussions

### Pix2Pix GAN results

The output z-stacks of the *Pix2Pix* GAN for the 19 validation z-stacks of the clean dataset were manually evaluated to determine the accuracy with which the network can predict the three-dimensional locations of mitochondrial fission, fusion and depolarisation events. This was done by comparing the predicted events to the relevant frames of the time-lapse sequence. The 19 validation z-stacks of the clean dataset were of a high quality and it was decided that using the low quality validation z-stacks from the complete dataset would not be an accurate indication of the network's event location prediction capabilities, because MEL also performed poorly on these z-stacks.

The total number of predicted and correctly predicted fission, fusion and depolarisation events are summarised in Table 6. The number of predicted and correctly predicted events for each of the validation images is provided in S3 Fig.

**Table 6. The location prediction accuracies of the three-dimensional *Pix2Pix* GAN for mitochondrial fission, fusion and depolarisation events.**

| Event type | Predicted events | Correctly predicted events | Accuracy |
|---|---|---|---|
| *Fission* | 181 | 65 | 35.9% |
| *Fusion* | 193 | 64 | 33.2% |
| *Depolarisation* | 163 | 8 | 4.90% |

### Discussion and conclusion regarding the three-dimensional Pix2Pix generative adversarial network

A three-dimensional variation of the *Pix2Pix* GAN was trained to predict the locations of mitochondrial fusion, fission and depolarisation events. This implementation achieved accuracies of 35.9% for fusion, 33.2% for fission and 4.90% for depolarisation events, as was shown in Table 6.

Low accuracies were expected since a relatively small dataset was used to train the network. The ground truth data generated with MEL is not an absolute ground-truth and therefor it is expected that a larger and more accurate dataset will yield better results.

It is also possible, however, unlikely that these percentages are the upper regions of accuracy when attempting to predict the three-dimensional locations of mitochondrial fission, fusion and depolarisation events with a single z-stack as input.

### Vox2Vox generative adversarial network results

As for the *Pix2Pix* GAN, the accuracy of the *Vox2Vox* GAN, for the location prediction of mitochondrial fission, fusion and depolarisation events, was determined by manually evaluating the results of the network for the 19 validation z-stacks of the clean dataset. The total number of predicted and correctly predicted events are listed in Table 7.

### Discussion and conclusion regarding the results of the Vox2Vox GAN

The location prediction accuracies of the *Vox2Vox* GAN were 37.1%, 37.3% and 7.43%, respectively for fission, fusion and depolarisation events, as was listed in Table 7. The number of predicted and correctly predicted events for each of the validation images is provided in S4 Fig.

The *Vox2Vox* GAN also achieved better accuracies than the three-dimensional *Pix2Pix* GAN. The *Vox2Vox* GAN achieved this regardless of being a shallower network, which could adversely affect the accuracy of its predictions. This improvement could be ascribed to the *ResNet* blocks, which are used in the bottle-neck of the generator architecture. These blocks passed additional information through the bottle-neck via the short skip connections.

## Comparison of results with mitochondrial event localiser accuracies

MEL was used to generate the ground-truth data that was used to train the *Pix2Pix* and *Vox2Vox* GANs. MEL achieved accuracies of 41.0%, 38.0% and 10.0% for fission, fusion and depolarisation events, respectively, for the dataset used in this paper. These accuracies were obtained by validating the events that were localised by MEL.

The *Pix2Pix* and *Vox2Vox* GANs (Tables 6 and 7) achieved comparable accuracies and the exceptionally low depolarisation event localisation accuracy of MEL explains the low accuracies of the GANs when predicting the locations of these events.

The GANs made their predictions using a single frame of a time-lapse sequence as input, in contrast to the two successive frames of a time-lapse sequence required as input for MEL.

**Table 7. The location prediction accuracies of the three-dimensional *Vox2Vox* GAN for mitochondrial fission, fusion and depolarisation events.**

| Event type | Predicted events | Correctly predicted events | Accuracy |
|---|---|---|---|
| *Fission* | 213 | 79 | 37.1% |
| *Fusion* | 233 | 87 | 37.3% |
| *Depolarisation* | 148 | 11 | 7.43% |

Notably, the GANs required less input information by predicting mitochondrial events using the information encoded only in the morphology of the mitochondria to achieve accuracies that are comparable to those achieved by MEL. Furthermore, the GANs made these predictions at a fraction of the time MEL requires to process the images. Neural networks have an inference time of a few seconds whereas MEL took 3–4 minutes to process a pair of z-stack images.

## Conclusion

In this paper a modified version of the *Pix2Pix* and *Vox2Vox* GANs were trained to predict the three-dimensional locations of mitochondrial fission, fusion and depolarisation events using a *single frame* of a time-lapse sequence as input. The training dataset was generated using the MEL method.

The *Pix2Pix* GAN was capable of predicting the locations of mitochondrial fission, fusion and depolarisation events with an accuracies of 35.9%, 33.2% and 4.90%, respectively. Similarly, the *Vox2Vox* GAN was capable of predicting the locations of these events, with accuracies of 37.1%, 37.3% and 7.43%. The *Vox2Vox* GAN therefor outperformed the *Pix2Pix* GAN for the task of three-dimensional mitochondrial event location prediction.

The low prediction accuracies, although comparable to the un-validated results achieved by MEL, can be ascribed to inaccuracies in the dataset that was used for training of the neural networks. It is expected that these location prediction accuracies will improve with a larger and more accurate dataset. However, it may be possible that the accuracies obtained by the neural networks were the highest accuracies that can be achieved with these methods.

The location prediction accuracies that were obtained are currently too low for a viable implementation of these tools in the life sciences industry. However, it is clear that the morphology of the mitochondria were modelled by the networks to some degree of accuracy. These networks can be used for a qualitative study on where regions of mitochondria are likely to undergo events.

The use of deep neural networks to predict the occurrence and exact location of mitochondrial fission, fusion and depolarisation events in three-dimensions, has, to our knowledge, never been achieved before. For this reason, fission and fusion event location prediction accuracies in excess of 30% is acceptable. Future research projects can use these results as baseline for their results.

## Supporting information

**S1 Fig. Possible causes of incorrect event localisations by MEL.** Two frames (a and b) of a time-lapse sequence show structures which would incorrectly be classified by MEL as fusion candidates, highlighted with red circles. The thresholded versions of these two frames are shown in (c) and (d). MEL would incorrectly localise this event due to the binarisation of the low intensity bridge between the structures.
(TIF)

**S2 Fig. The post MEL process validation tool.** An image of the post MEL process event validation tool used to validate the legitimacy of events, where a fission event was analysed. In the left panel the two structures in Frame2 that underwent fission are shown in red and green. The deconvolved Frames1 and 2 are then shown to assist the user in determining whether the event is real. The panel on the right shows a three-dimensional model of the red and green structures in Frame2 overlayed on the original structure in Frame1 (grey structure). The

fission event is indicated by the red dot. Reproduced form Theart *et al.,* *[18]*.
(TIF)

**S3 Fig. The number of predicted events by the *Pix2Pix* GAN.** Predicted and correctly predicted mitochondrial fusion (a), fission (b) and depolarisation (c) events by the *Pix2Pix* GAN for the 19 validation images used during training.
(TIF)

**S4 Fig. The number of predicted events by the *Vox2Vox* GAN.** Predicted and correctly predicted mitochondrial fusion (a), fission (b) and depolarisation (c) events by the *Vox2Vox* GAN for the 19 validation images used during training.
(TIF)

## Author Contributions

**Conceptualization:** Rensu P. Theart.

**Formal analysis:** James G. de Villiers.

**Investigation:** James G. de Villiers.

**Methodology:** James G. de Villiers.

**Project administration:** James G. de Villiers.

**Software:** James G. de Villiers.

**Supervision:** Rensu P. Theart.

**Validation:** James G. de Villiers.

**Visualization:** James G. de Villiers.

**Writing – original draft:** James G. de Villiers.

**Writing – review & editing:** James G. de Villiers, Rensu P. Theart.

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
