## [Decision Letter · Decision Letter 0]

11 Oct 2022

PONE-D-22-18015Three-dimensional mitochondrial fission, fusion and depolarisation event location prediction for a high throughput analysis of fluorescence microscopy imagesPLOS ONE

Dear Dr. Theart,

Thank you for submitting your manuscript to PLOS ONE. After careful consideration, we feel that it has merit but does not fully meet PLOS ONE’s publication criteria as it currently stands. Therefore, we invite you to submit a revised version of the manuscript that addresses the points raised during the review process.

We look forward to receiving your revised manuscript.

Kind regards,

Xiyu Liu

Academic Editor

PLOS ONE

Journal Requirements:

The financial assistance of the National Research Foundation (NRF) towards this 406

research is hereby acknowledged (grant MND200423516047).

However, funding information should not appear in the Acknowledgments section or other areas of your manuscript. We will only publish funding information present in the Funding Statement section of the online submission form. 

- JG de Villiers was received a scholarship from the National Research Foundation (NRF). 

- Grant number: MND200423516047

- National Research Foundation (NRF)

- URL: https://www.nrf.ac.za/

- No, they played no role in the study. 

The co-author, Dr RP Theart, is the author and creator of the Mitochondrial Event Localiser, which was used to generate the ground-truth data for this project. His research is also published in PLOS ONE (https://doi.org/10.1371/journal.pone.0229634). 

Reviewers' comments:

Reviewer's Responses to Questions

**Comments to the Author**

1. Is the manuscript technically sound, and do the data support the conclusions?

Reviewer #1: Partly

Reviewer #2: Partly

2. Has the statistical analysis been performed appropriately and rigorously? 

Reviewer #1: Yes

Reviewer #2: Yes

3. Have the authors made all data underlying the findings in their manuscript fully available?

Reviewer #1: Yes

Reviewer #2: No

4. Is the manuscript presented in an intelligible fashion and written in standard English?

Reviewer #1: Yes

Reviewer #2: Yes

5. Review Comments to the Author

Reviewer #1: In the paper the prediction of occurrence and exact locations of mitochondrial fission, fusion and depolarisation events in three dimensions by using two types of a generative adversarial network (GAN) is presented. The strength of the paper is the innovative application of the GAN network to the aforementioned prediction. On the other hand, the weakness of the work is that both types of applied GAN networks achieved worse results than the MEL used so far. Nevertheless, due to the innovative approach to the problem, as well as the prospect of improving the described approach, the journal may be published in PLOS One.

Reviewer #2: This work is quite interesting; however, I have various issues which are given below.

1. The authors have not mentioned what’s the motivation and challenges. Such as “This paper documents the development of a novel method to predict the occurrence and exact locations of mitochondrial fission, fusion and depolarisation events in three dimensions” But why?

2. In this paper the author used two types of network. Such as “These occurrence and location of these events were successfully predicted with a three-dimensional version of the Pix2Pix generative adversarial network (GAN) as well as a three-dimensional adversarial segmentation network called the Vox2Vox GAN.”. Here again a question rises that why the authors used this network.

3. I have not found any details of methodology in introduction. Briefly describe the methodology along with motivation, challenges and related work.

4. Related work section shows a detailed story of various existing methods. The authors should connect these stories with the proposed work. Second, why a separate section for each method?

5. The heading “Event location penalised MAE loss (ELP-MAE)” is quite confusing. Please enhance it.

6. It is better not to use abbreviations in the heading. “Comparison of results with MEL accuracies”. Also, the author should compare the proposed with atleast three exiting methods.

7. The title of the paper is very ambiguous. Use a more concise title.

6. PLOS authors have the option to publish the peer review history of their article (what does this mean?). If published, this will include your full peer review and any attached files.

Reviewer #1: No

Reviewer #2: No

---

## [Author Response · Author response to Decision Letter 0]

24 Nov 2022

Response to Reviewer 1 Comment 1: I would like to thank the reviewer for pointing out ambiguity in the submission. The MEL method receives two sequential frames of a time-lapse sequence as input. It then compares these two frames to determine the locations of mitochondrial fission, fusion and depolarisation events. Therefore, MEL localises these events based on two frames. The GANs discussed in this submission, on the other hand, receives only a single frame of a time-lapse sequence as input and is tasked with predicting the locations of mitochondrial events. Consequently, the GANs have far less information at their disposal and consequently, the fact that they achieved comparable accuracy to the MEL method is a very positive results indicating that the current morphology of the mitochondria contains a significantly amount of information to aid this prediction. In an effort to reduce this ambiguity a specific reference to MEL using two sequential frames of a time-lapse sequence as input has been added to the introduction and related work sections of the submission.

Response to Reviewer 2 Comment 2: I appreciate the reviewer pointing out this lack of information in the submission. The following paragraph was added to the abstract: "This novel implementation of neural networks to predict these events using information encoded only in the morphology of the mitochondria eliminate the need for time-lapse sequences of cells. The ability to predict these morphological mitochondrial events using a single image can not only democratise research but also revolutionise drug trials."

Response to Reviewer 2 Comment 2: I would like to thank the reviewer for this comment. In an attempt to explain the use of these networks the following paragraph was added to the introduction: "The ability to predict the locations of these events on a z-stack image using only a single image and information encoded only in the morphology of the mitochondria implied that generative networks were needed. For this reason generative neural networks were used. These networks were trained using pairs of input and desired output images. To our knowledge the use of generative networks to predict the locations of mitochondrial events has never been done before."

Response to Reviewer 2 Comment 3: I would like to thank the reviewer for pointing out this omission in the submission. I have added the following to my Introduction: "Until recently, the occurrence and locations of these events were determined through manual visual inspection by comparing two frames of a time-lapse sequence. Recently, an automatic method for comparing two frames of a time-lapse sequence to localise these events has been developed. This automatic method was used to generate the training data for the neural networks discussed in this paper. These networks use a single frame of a time-lapse sequence as input and makes predictions using information encoded in the morphology of the mitochondria.

Mitochondrial event predictions using neural networks enable insight into the health of cells from a single z-stack, thereby eliminating the need to capture time-lapse sequences. The removal of time-lapse sequences from the analysis process enables the use of fixed cells rather than live cells while still obtaining similar analysis insights. Furthermore, neural networks democratise this type of research by providing researchers who have no access to the equipment necessary to obtain time-lapse sequences with the opportunity to conduct similar studies."

Response to Reviewer 2 Comment 4: I would like to thank the reviewer for pointing out this error in my submission The two GANs are now discussed in the same section and the following paragraph was added at the start of the section: "To our knowledge, no other methods capable of predicting these events exist and as a consequence this section discusses the automatic method that was is used for localising mitochondrial fission, fusion and depolarisation events by comparing two frames of a time-lapse sequence, which was used to generate the ground truth data for the neural networks discussed in this paper. This section also discusses the most prominent generative neural networks used in industry."

Response to Reviewer 2 Comment 5: I would like to thank the reviewer for pointing out this point of confusion in the submission. The heading of the section has been changed to the following: "Location based weighted mean absolute error loss function"

Response to Reviewer 2 Comment 6: I would like to thank the reviewer for this comment. All abbreviations were removed from the headings.

Unfortunately, MEL is to our knowledge the only method currently capable of automatically localising mitochondrial events in three dimensions. Furthermore, the use of GANs to predict the locations of these events are also, to our knowledge, the first of their kind and we can thus not compare them to any other methods.

Response to Reviewer 2 Comment 7: I would like to thank the reviewer for this comment. The title has been changed to the following: "Predicting mitochondrial fission, fusion and depolarisation event locations from a single z-stack"

---

## [Decision Letter · Decision Letter 1]

2 Jan 2023

PONE-D-22-18015R1Predicting mitochondrial fission, fusion and depolarisation event locations from a single z-stackPLOS ONE

Dear Dr. Theart,

Thank you for submitting your manuscript to PLOS ONE. After careful consideration, we feel that it has merit but does not fully meet PLOS ONE’s publication criteria as it currently stands. Therefore, we invite you to submit a revised version of the manuscript that addresses the points raised during the review process.

We look forward to receiving your revised manuscript.

Kind regards,

Xiyu Liu

Academic Editor

PLOS ONE

Journal Requirements:

Reviewers' comments:

Reviewer's Responses to Questions

**Comments to the Author**

1. If the authors have adequately addressed your comments raised in a previous round of review and you feel that this manuscript is now acceptable for publication, you may indicate that here to bypass the “Comments to the Author” section, enter your conflict of interest statement in the “Confidential to Editor” section, and submit your "Accept" recommendation.

Reviewer #1: All comments have been addressed

Reviewer #3: All comments have been addressed

2. Is the manuscript technically sound, and do the data support the conclusions?

Reviewer #1: Yes

Reviewer #3: Partly

3. Has the statistical analysis been performed appropriately and rigorously? 

Reviewer #1: N/A

Reviewer #3: Yes

4. Have the authors made all data underlying the findings in their manuscript fully available?

Reviewer #1: Yes

Reviewer #3: Yes

5. Is the manuscript presented in an intelligible fashion and written in standard English?

Reviewer #1: Yes

Reviewer #3: Yes

6. Review Comments to the Author

Reviewer #1: (No Response)

Reviewer #3: The author addressed the comments from the previous reviewers. There are some concerns remaining.

1. TMRE is membrane potential dependent chemical dye. The manuscript shall address this issue in the discussion. If the authors could apply membrane potential insensitive dye or fluorescence protein targeting to mitochondria to perform the prediction under different stress condition (with/without CCCP), the readers will be much more appreciated.

2. The cause-effect relation between mtDNA and mitophagy described in line 7-8 is controversy.

3. The description of membrane depolarization and ROS in line 18-19 is also controversy.

7. PLOS authors have the option to publish the peer review history of their article (what does this mean?). If published, this will include your full peer review and any attached files.

Reviewer #1: No

Reviewer #3: No

---

## [Author Response · Author response to Decision Letter 1]

13 Feb 2023

Comment 1: 

TMRE is membrane potential dependent chemical dye. The manuscript shall address this issue in the discussion. If the authors could apply membrane potential insensitive dye or fluorescence protein targeting to mitochondria to perform the prediction under different stress condition (with/without CCCP), the readers will be much more appreciated.

Response 1:

We would like to thank the reviewer for pointing out this possible source of gaining better reader satisfaction.

We have now updated the the Materials and Methods section to clarify our justification for using TMRE, by stating "TMRE is used widely when assessing mitochondrial dysfunction and has a good signal-to-noise ratio. Additionally, since TMRE is a membrane potential dependent chemical dye it allows us to create the ground truth data for mitochondrial depolarisation which MEL is able to detect." Furthermore, since the MEL method, on which this new work is based, also used TMRE in the development, it was deemed to provide the most direct comparison with that method. However, since we argue that our method primarily rely on the morphology of the mitochondrial network, future work should be able to use other mitochondrial probes, such as MitoTracker, and achieve similar result.

Comment 2:

The cause-effect relation between mtDNA and mitophagy described in line 7-8 is controversy.

Response 2:

We appreciate the reviewer pointing out this point of controversy in the article. 

Indeed the link between mitochondrial events and mitophagy is controversial. However, there are articles documenting a link between mtDNA and mitophagy. References to these articles have been added to the article:

"To reduce the risk of mtDNA mutation accumulation, a mitochondrion can undergo dynamic events such as fission, fusion and mitochondrial autophagy (mitophagy) [Parone et al., Mourier et al.]."

Furthermore, since this is only provided for some context and motivation for the study in this paper, and not part of the main study itself, no additional changes was deemed necessary.

Comment 3:

The description of membrane depolarization and ROS in line 18-19 is also controversy.

We appreciate the reviewer for pointing out this point of controversy in the article. We have clarified this as shown below. Furthermore, since this is only provided for some context and motivation for the study in this paper, and not part of the main study itself, no additional changes was deemed necessary.

The following was added to to the Introduction:

The mitochondrial respiratory chain is the main site of cellular ROS production, while mitochondria are also highly sensitive to, and affected by, ROS toxicity [Wu et al.]. ROS induced ROS release (RIRR) has been described as a positive feedback mechanism that is involved in the interaction between ROS and the mitochondria where a surge of mitochondrial ROS is induced by ROS, subsequently causing a reduction in the mitochondrial membrane potential [Fang et al.]. For this reason mitochondrial depolarisation is required for mitochondrial quality control and removal of dysfunctional mitochondria through mitophagy.

---

## [Decision Letter · Decision Letter 2]

20 Feb 2023

Predicting mitochondrial fission, fusion and depolarisation event locations from a single z-stack

PONE-D-22-18015R2

Dear Dr. Theart,

We’re pleased to inform you that your manuscript has been judged scientifically suitable for publication and will be formally accepted for publication once it meets all outstanding technical requirements.

Kind regards,

Xiyu Liu

Academic Editor

PLOS ONE

Additional Editor Comments (optional):

Reviewers' comments:

Reviewer's Responses to Questions

**Comments to the Author**

1. If the authors have adequately addressed your comments raised in a previous round of review and you feel that this manuscript is now acceptable for publication, you may indicate that here to bypass the “Comments to the Author” section, enter your conflict of interest statement in the “Confidential to Editor” section, and submit your "Accept" recommendation.

Reviewer #3: All comments have been addressed

2. Is the manuscript technically sound, and do the data support the conclusions?

Reviewer #3: (No Response)

3. Has the statistical analysis been performed appropriately and rigorously? 

Reviewer #3: (No Response)

4. Have the authors made all data underlying the findings in their manuscript fully available?

Reviewer #3: (No Response)

5. Is the manuscript presented in an intelligible fashion and written in standard English?

Reviewer #3: (No Response)

6. Review Comments to the Author

Reviewer #3: (No Response)

7. PLOS authors have the option to publish the peer review history of their article (what does this mean?). If published, this will include your full peer review and any attached files.

Reviewer #3: No

---

## [Editor Report · Acceptance letter]

27 Feb 2023

PONE-D-22-18015R2 

Predicting mitochondrial fission, fusion and depolarisation
event locations from a single z-stack 

Dear Dr. Theart:

I'm pleased to inform you that your manuscript has been deemed suitable for publication in PLOS ONE. Congratulations! Your manuscript is now with our production department. 

Kind regards, 

on behalf of

Professor Xiyu Liu 

Academic Editor

PLOS ONE